# INTERPRETABLE CONVOLUTIONAL FILTER PRUNING

## ABSTRACT

The sophisticated structure of Convolutional Neural Network (CNN) allows for outstanding performance, but at the cost of intensive computation. As significant redundancies inevitably present in such a structure, many works have been proposed to prune the convolutional filters for computation cost reduction. Although extremely effective, most works are based only on quantitative characteristics of the convolutional filters, and highly overlook the qualitative interpretation of individual filter's specific functionality. In this work, we interpreted the functionality and redundancy of the convolutional filters from different perspectives, and proposed a functionality-oriented filter pruning method. With extensive experiment results, we proved the convolutional filters' qualitative significance regardless of magnitude, demonstrated significant neural network redundancy due to repetitive filter functions, and analyzed the filter functionality defection under inappropriate retraining process. Such an interpretable pruning approach not only offers outstanding computation cost optimization over previous filter pruning methods, but also interprets filter pruning process.

## 1 INTRODUCTION

The great success of Convolutional Neural Network (CNN) is benefited from its advanced algorithm and architecture, which utilize inter-connected multi-layer network structure to hierarchically abstract the data feature for recognition tasks Krizhevsky & *et al* (2012). However, this complex mechanism offers CNN outstanding performance at the price of intensive computation cost. Therefore, many CNN optimization works have been proposed to alleviate this computation cost Han & *et al* (2015), Jaderberg & *et al* (2014), *etc*.

While those network redundancy leveraging works are mostly based on quantitative evaluation, the qualitative analysis based on network interpretation is highly overlooked. With the neural network interpretation, CNN can be well analyzed regarding its hierarchical structure and individual component's functionality, and therefore no longer a "black-box". However, most neural network interpretation works is merely considered as a post-analysis approach, and very few works have utilized it to analysis and guide the optimization directly (Yosinski et al. (2015)).

Considering the absence of the network interpretation analysis and optimization, in this work we utilized different CNN interpretation techniques to analysis the convolutional filter functionality and optimize the filter pruning method. In this work, we have the following major contributions:

- We utilized CNN visualization techniques and backward-propagation gradients analysis to interpret the convolutional filter functionality, and demonstrate significant network redundancy due to repetitive filter functions rather than insignificant magnitude;

- We designed a hierarchical filter functionality-oriented pruning method to explore the interpretable neural network optimization;

- We observed the filter functionality transission with model tuning, and reveals the inappropriate pruning and retraining methods may significantly defect the filter functionality and cause unnecessary network reconstruction effort.

Experiment results with various CNN models and image dataset show that, the proposed filter interpretation approach can effectively analysis the convolutional filters' the functionality and similarity. The proposed functionality-oriented pruning methods also achieves outstanding performance compared to traditional filter pruning methods with better training efficiency and interpretability

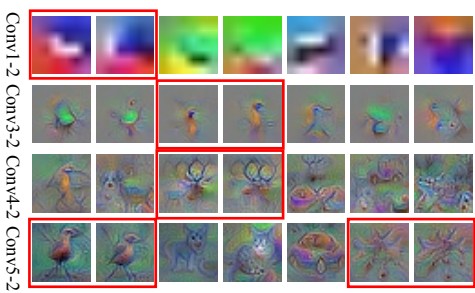

Figure 1: The illustration of visualized patterns of the convolutional filters.

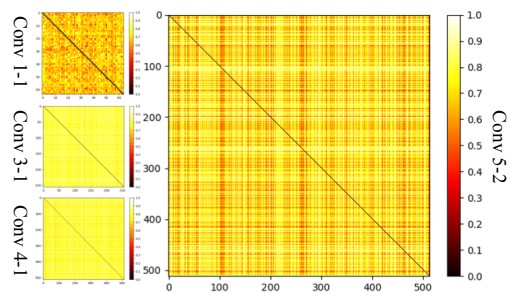

Figure 2: Filter similarity matrices.

## 2 RELATED WORK

**Convolutional Filter Pruning**  It is well known that the major computation cost in a CNN comes from the convolutional layers. Therefore, most CNN optimization works fall into convolutional filter pruning or compact CNN training with convolutional filter constraints: Li et al. (2016) ranked the convolutional filters with their absolute weight sum as the filter significance indication and pruned the insignificant filters for better computation efficiency, which is well-recognized and $\ell_1$ ranking based prunning; Hu et al. (2016) evaluated the filter significance calculating the occupation of zero valued weights for pruning guidance; He et al. (2017) also applied Lasso-regression in the filter pruning for batched processing. The filter significance is also evaluated with the filter feature maps: Polyak & Wolf (2015) pruned the filters with small feature map values; Luo et al. (2017) evaluate filter's significance by testing its impact on feature map reconstruction errors after being pruned.

Most of these works are based on quantitative analysis on the filter significance. However, such an approach is already questioned by some recent works: Huang et al. (2018) proved that certain redundancy also exists in filters with large "significance"; while Ye et al. (2018) correspondingly demonstrated that filters with small values also significantly affect the neural network performance. Therefore, rather than merely evaluating the convolutional filters in a quantitative approach, we analysis the filter functionality in a qualitative manner to guide the neural network optimization.

**Convolutional Filter Visualization**  As the convolutional filters are designed to capture certain input features, the semantics of the captured feature can effectively indicate the functionality of each filter. However, as the convolutional filter are embodied by matrices, the functionality is hard to directly interpret, which significantly hinders the qualitative neural network optimization.

Recently, many CNN visualization techniques have been proposed to qualitatively analysis the CNN in perspectives of network structure and component functionality, such as Activation Maximization (AM) (Yosinski et al. (2015)), Network Inversion (Mahendran & Vedaldi (2015)), Deconvolutional Neural Networks (Zeiler & Fergus (2014)), and Network Dissection based Visualization (Zhou et al. (2016)). Among those methods, AM offers the most efficient and effective interpretation for the convolutional filter functionality, which is defined by each filter's maximum activation pattern. In this work, we utilize the AM as our major visualization tool for convolutional filter analysis.

## 3 CONVOLUTIONAL FILTER FUNCTIONALITY INTERPRETATION AND FILTER REDUNDANCY ANALYSIS

Different from previous quantitative filter significance analysis, the convolutional filter functionality is qualitatively interpreted in this paper. Based on the functionality interpretation, the neural network redundancy is further analyzed regarding the filter functionality. For preliminary demonstration, we adopt a VGG-16 model trained on CIFAR-10 dataset for intuitive result demonstration.

## 3.1 Interpretable Convolutional Filter Functionality Analysis

To qualitatively interpret the convolutional filter functionality, both AM visualization and backward-propagation gradients analysis are utilized. AM visualization interprets a filter's feature extraction preference from the neural network inputs, while the backward-propagation gradients evaluate the filters' contribution to classification outputs.

In AM visualization, the feature extraction preference of a filter $F_i^l$ from layer $L_l$ is represented by a synthesized input image $X$ that causes the highest activation of $F_i^l$ (*i.e.* the convolutional feature map value). Mathematically, the synthesis process of such an input can be formulated as:

$$V(F_i^l) = \arg\max_X A_i^l(X), \qquad X \leftarrow X + \eta \cdot \frac{\partial A_i^l(X)}{\partial X}, \tag{1}$$

where $A_i^l(X)$ is the activation of filter $F_i^l$ from an input image $X$, $\eta$ is the gradient ascent step size. With $X$ initialized as an input image of random noises, each pixel of this input is iteratively changed along the $\partial A_i^l(X)/\partial X$ increment direction to achieve the highest activation. Eventually, $X$ demonstrates a specific visualized pattern $V(F_i^l)$, which contains the filter's most sensitive input features with certain semantics, and represents the filter's functional preference for feature extraction.

Fig. 1 demonstrates several filters' visualized patterns from different layers. By interpreting the content of these patterns, it is clear to see that the filters in lower layers are more sensitive to fundamental colors and shapes. While with the layer depth increasing, the patterns tend to show specific objects and eventually recognizable class targets.

In the backward-propagation gradients analysis, the gradients indicate the impact of each convolutional filter to individual classification targets:

$$\gamma_{i,y_j} = \frac{1}{N} \sum_{n=1}^{N} \left\| \frac{\partial P(y_j)}{\partial A_i^l(x_n)} \right\|, \tag{2}$$

where $P(y_j)$ is the output probability of a sample image $x_n$ to the class $y_j$, and $A_i^l(x_n)$ is the activation of filter $F_i^l$ for each test image $x_n$. $\partial P(y_j)/\partial A_i(x_n)$ is the backward-propagation gradient of class $y_j$ respective to the filter's activation, which indicates the dedicated contribution of filter $F_i^l$ to class $y_j$. With preliminary experiments, we also find that some filters' visualized patterns also constrain obvious class objects, which indicates large impact to certain classes $y_j$. Therefore, the connection between filters and dedicated class outputs can be also determined.

Associating these two analysis methods, namely the CNN visualization and backward-propagation gradients analysis, the filters' functionalities can be well interpreted in a qualitative manner.

## 3.2 Function Similarity based Filter Redundancy Analysis

Based on the above analysis, the convolutional filter functionality can be well interpreted regarding the input feature extraction preference and output contribution. These qualitative interpretations also guide certain neural network redundancy analysis, where filters with similar functionalities may indicate unnecessary structural repetition. Such repetition can be well identified among the filters with similar visualized patterns, as shown in the Fig. 1: as denoted by the red blocks, significant functionality similarities present among the convolutional filters in each layer.

To quantitatively determinate possible filter functionality repetition, a similarity degree between filter $F_k^{(c,l)}$ and $F_i^{(c,l)}$ is formulated regarding both AM visualized patterns and the back-propagation gradients:

$$S_D[V(F_i^{(c,l)}), V(F_k^{(c,l)})] = \|V(F_i^{(c,l)}) - V(F_k^{(c,l)})\|^2 \cdot \|\gamma_i - \gamma_k\|^2, \tag{3}$$

where $\|V(F_i^{(c,l)}) - V(F_k^{(c,l)})\|^2$ calculates the Euclidean distance between two visualized patterns, which indicate feature extraction preference similarity [1]. And $\|\gamma_i - \gamma_k\|^2$ calculates the similarity between the two filter's contribution to specific output classes. To equally evaluate these two criterion, both vectors are normalized, and the summation is utilized as the filter similarity degree.

---

[1]Although image similarity analysis tools like SSIM are more specialized, they cannot be applied into low layer filters with extremely small resolutions. Therefore, the Euclidean distance is adopted for its generality.

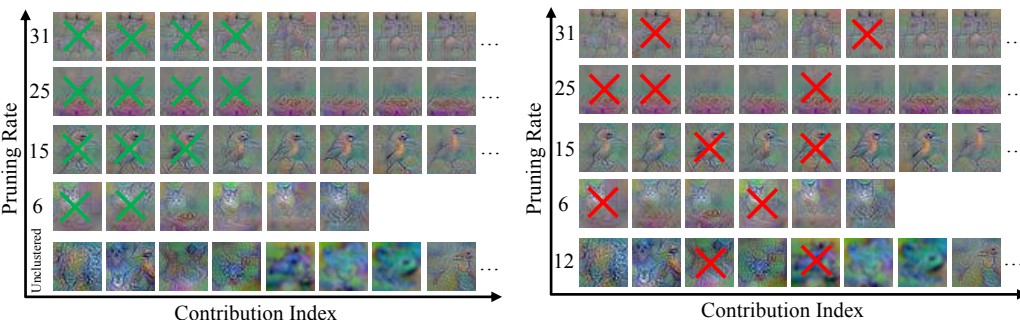

(a) Proposed functionality-oriented filter pruning.     (b) $\ell_1$ ranking based filter pruning.

Figure 3: Case study of the hierarchical filter functionality-oriented and filter $\ell_1$ ranking based filter pruning on the Conv5_1 of VGG-16. The convolutional filters are shown by their visualized patterns, and aligned according to the visualization based filter clustering.

According to the qualitative analysis of Eq.3, a set of matrices filter similarity degrees are constructed as shown in Fig.2. The matrix axes represent the filter index in each layer (*e.g.* 512x512 for filter Conv5_2), and each cross-point indicates the similarity degree $S_D$ between two filters. The magnitudes of the similarity degrees are denoted by a color range, where the minimum is denoted as red and the maximum is denoted as white. From Fig.2, it can be observed that the filter similarity widely present in different layers. Meanwhile, significant filter similarity exit in the shallowest and deepest layers of the neural network (*e.g.* Conv1_1 and Conv5_2). Those high similarity degrees indicate considerable functionality repetition among the convolutional filters, which might be leveraged for effective neural network optimization.

In later sections, the filter functionality evaluation approach is widely applied for network redundancy analysis and filter pruning guidance.

## 4 HIERARCHICAL FILTER FUNCTIONALITY-ORIENTED PRUNING

Based on the convolutional filter functionality and redundancy analysis, we further propose a novel hierarchical filter pruning method oriented by the qualitative functionality interpretation.

### 4.1 PRUNING METHOD OVERVIEW

The proposed functionality-oriented filter pruning method is designed in a hierarchical manner to address the functionality redundancy in different neural network component levels.

The method has the following major stages: (1) Filter Clustering – Based on the aforementioned filter similarity analysis with AM visualization and the backpropagation gradients, the filters with similar functionalities in each layer are clustered into multiple groups. The filters in the same cluster are considered to have repetitive functionalities; (2) Filter Level Pruning – Inside each cluster, the filters' relative contribution to the output classes are qualitative evaluated by the backpropagation gradients analysis. The contribution index of each filter can be considered as the filter significance regarding its functionality; (3) Cluster Level Pruning – Inside each layer, the relative cluster pruning rate is based on the cluster volume size to guide the filter pruning in each cluster; (4) Layer Level Pruning – Guided by the layer sensitive to pruning, the relative pruning ratio for each layer is calculated to guide the pruning in each layer and therefore each cluster.

Fig. 3(a) illustrate an intuitive example for the filter clustering in the layer Conv5_1, where each row represents one cluster in the layer, and the horizontal axis indicates the filter's contribution index. Given a pruning rate for the model, the relative pruning rate for each layer and cluster can be determined, and the filters with least classification contribution will be pruned first (marked by green crosses). Meanwhile, we also demonstrate $\ell_1$ ranking based filter pruning in the Fig. 3(b). Considering $\ell_1$ ranking's quantitative filter significance evaluation, the pruned filters (marked by red crosses) have no interpretable selection patterns and no functionality correlations. Moreover,

---

**Algorithm 1** Visualization based k-means Filter Clustering

---

**Input:** CNN, Number of layers L, and Number of filters in each layer $I_l$
**Output:** L Clusters
 1: **for each** $Layer\ L_l$ **do** $List_l$=[];
 2:     **for each** $Filter\ F_i^l$ **do** Generate the pattern: $V(F_i^l)$=$X_i^l$; $List_l$.append($V(F_i^l)$);
 3: **for** c in range (1, $I_l/2$) **do** Grid search all possible cluster numbers: $C_l$=kmeans.cluster($List_l$, c);
 4:     **for** $C_l^c$ in $C_l$ **do** Merge the single filter clusters: $C_{locked}^l$ = [];
 5:         **if** Len($C_c^l$)== 1 **then** $C_{locked}^l$.append($C_c^l$); k=k-1.
 6: Select the maximal c as the optimal cluster number: c=Max(c) **return** $C_c^l$

---

comparing to $\ell_1$ ranking based filter pruning, the proposed functionality-oriented pruning requires extremely small effort for model retraining, which will be further explained in later sections.

In the next section, the details for each processing stage will be presented.

## 4.2 HIERARCHICAL FILTER PRUNING SCHEME

**Filter Clustering**   $S_D$ derived from Eq. 3 demonstrates the functionality similarity of any two convolutional filter. While to evaluate all the filters' similarity in a comprehensive neural network model, we applies *k*-means algorithm in each layer to cluster the filters.

As described in Alg. 1, we first utilize AM to obtain the visualized pattern of each convolutional filter. Then, we apply the *k*-means algorithm to $S_D$ values of all the convolutional filters in each individual layer. To determine proper cluster amount in each layer (i.e. *k*), a grid search is conducted from one to half of the total filter number. It is worth noting that there are inevitably certain amount of filters with extremely minimal similarity with others, which are also group together and won't be considered for pruning due to their possible instinct functionalities.

Each of the convolutional filter clusters can be considered as an interpretable functionality node. However, they can be also considered as neural network redundancy units that extract repetitive features from the inputs, and possibly defect the neural network computation efficiency.

**Filter Level Pruning**   Ideally, the filters with the exactly same functionality can be substituted to each other, and each cluster can reserve only one filter and prune out the rest. However, due to the complex mechanism of neural networks, the filters with similar functionalities may have distinct contribution to the classification results. Therefore a relative filter contribution evaluation inside the cluster is necessary to determine the pruning significance for each filter.

To quantitatively identify the filters' contribution variation, the backward-propagation gradients is utilized to rank the filters in each cluster. Given a layer $l$ with $I_l$ filters, the neural network output could be formulated by any layer $l$'s output feature maps in the following format:

$$Z(F, A^l) = F^L(...\ \alpha(F^{l+1} * A^l + b^{l+1})...) + b^L,\qquad(4)$$

where $F$ is the filter weights and bias in every layer. $A^l$ is the output feature maps (activation maps) of layer $l$. Due to the high complexity of $F$ and multiple layers' combination, an approximate function $Z$ is needed. Here we use $Z$'s first-order Taylor expansion for approximating:

$$Z(A^l + \Delta) = Z(A^l) + \frac{\partial Z(A^l)}{\partial A^l} \cdot \Delta.\qquad(5)$$

In our pruning method, when filter $i$ in the $l-1$ layer is pruned, the filter's output feature map is corresponding set to zero, i.e. changing the $i^t h$ dimension of $A^l$ to zero. Therefore, the influence on $Z$ can be qualitatively evaluated as:

$$\frac{\partial Z(A_i^l)}{\partial A_i^l} \cdot \Delta,\ where\ \Delta = A_i^l \to 0.\qquad(6)$$

Before pruning, each filter $F_i^{(c,l)}$ in the cluster $C_c^l$ of layer $l$ is firstly ranked by the contribution index, $I(F_i^{(c,l)})$, which is calculated by examining the backward-propagation gradients. Specifically,

$$I(F_i^{(c,l)}) = \frac{1}{N} \sum_{n=1}^{N} \left\| \frac{\partial Z(F, A^l)}{\partial A_i^l(x_n)} \right\|, \tag{7}$$

where the $Z(F, A^l)$ is the neural network output loss of a sample image $n$, and the $A_i^l(x_n)$ is the feature map of filter $i$ for each test image $n$.

As shown in Fig. 3, the filters in each cluster are ranked with contribution index. When a specific filter pruning amount is assigned to the cluster, filters with small contribution will be firstly pruned.

**Cluster Level Pruning** As filter clusters are supposed to bear certain inner redundancy, they can perform filter pruning independently. Hence a parallel pruning operation can be well executed in each cluster as shown in Fig. 3. However, the significant variance of filter distribution also presents in each cluster. In each layer, a few clusters may have significantly larger volumes, while there are also a certain amount of filters that can't be well clustered. Therefore, a dedicated cluster level scheme is necessary to balance the pruning rates between clusters. Considering the cluster volume, we define an adaptive pruning rate of $R_{clt}^{(c,l)} = length(C_c^l)$. The $length(C_c^l)$ calculates the cluster volume size, since larger cluster volume demonstrates more filter redundancy. And a larger $R_{clt}^{(c,l)}$ value will lead significantly more aggressive cluster level pruning. Therefore, we can see different filter pruning amounts between clusters in the Fig. 3.

**Layer Level Pruning** As the functionalities of each layer are relatively independent, all the layers can be pruned in a parallel manner with minor accuracy defection. In our pruning method, we derive a hyper-parameter $Pr$ based on each layers' accuracy sensitive to the pruning. Specifically, we prune filters in each layer independently and evaluate the pruned model's accuracy on the test set without retraining. $Pr$ is the neural network accuracy drop by pruning certain amount filters in each layer. Since each layer has different accuracy sensitivity to pruning, we empirically determine the same accuracy drop $Pr$ for all layers. By setting the hyper-parameter $Pr$, we can find a certain amount of filters $r$ to prune for each layer, corresponding amount of filters can be well assigned to each cluster as $r \cdot R_{clt}^{(c,l)}$. And in each cluster, the filters to be pruned can be well identified by the contribution index of $I(F_k^{(c,l)})$.

Leveraging the convolutional filters' qualitative functionality analysis, our proposed method is expected to leverage the neural network interpretability for more accurate redundant filter allocation, faster pruning speed, and optimal computation efficiency.

## 5 EXPERIMENTS

In this section, we will evaluate the functionality-oriented filter pruning method and testify the interpretable filter functionality that the method derived from.

### 5.1 EXPERIMENT SETUP

The visualization analysis and filter pruning method are implemented in Caffe environment (Jia et al. (2014)). The evaluation is performed on our designed ConvNet and VGG-16 on CIFAR-10, VGG-16 and ResNet-32 on ImageNet.

In evaluations on CIFAR-10, a data argumentation procedure is processed firstly through horizontal flip and random crop, and a 4 pixel padded training data set is generated. The whole training data is utilized in calculating the contribution index in Eq.7 for more accurate estimation. Our designed ConvNet is designed based on the classic AlexNet model, in which the convolutional filter has the size of 3x3 and the number as the same as the original model [2].

In evaluations on ImageNet, 10 classes images out of 1000 classes, namely ImageNet-10 is utilized in this work. And each class contains 1300 training images and 50 validation images. The VGG-16 and ResNet-32 on ImageNet-10 is pre-trained with a batch size of 128 and a learning rate of $1e^{-3}$,

---

[2]The detailed neural network structure of the ConvNet and the VGG-16 are demonstrated in the Appendix.

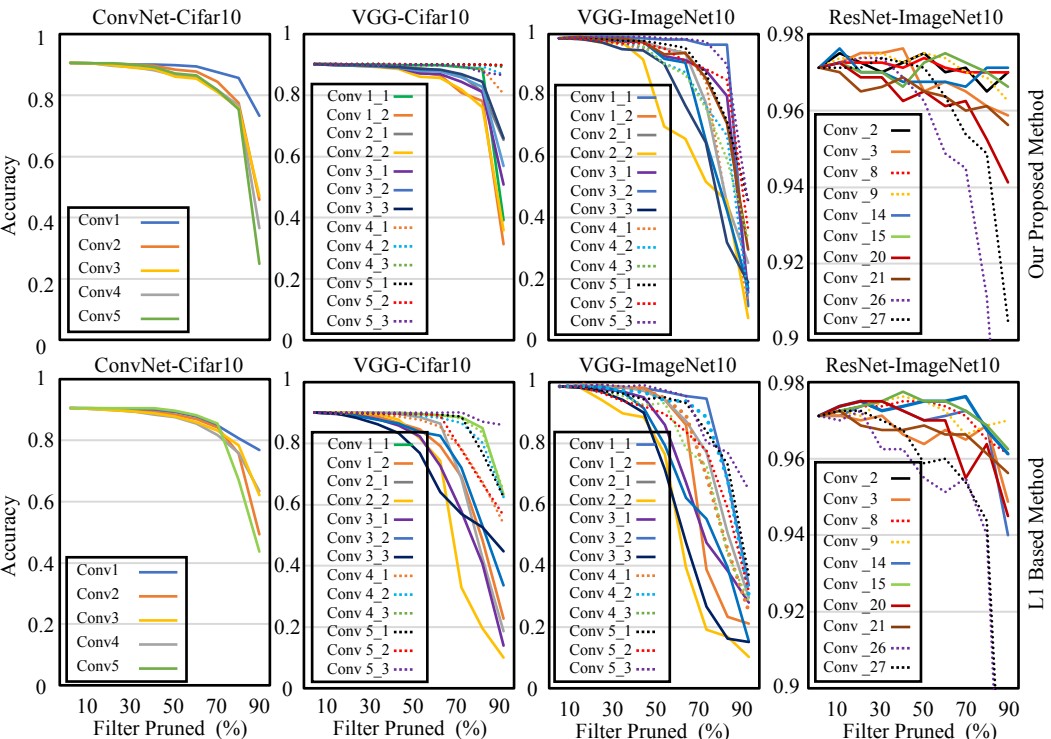

Figure 4: Sensitivity of each individual layer to pruning methods.

achieving 98.6% and 97.1% testing accuracy respectively. As the same, the train data set is fully utilized to calculate the contribution index.

The ResNets-32 for ImageNet have three stages of residual blocks for feature maps with sizes of 56x56, 28x28, and 14x14. The three stages contain 3, 4, and 2 residual blocks respectively, and each residual block has three convolutional layers. The first convolutional layer of ResNets-32 is numbered as 1 and the layer number is defined to increase from shallower to deeper layers. Note that the projection layer located in the junction of two adjacent stages is neither numbered nor pruned as was done in other previous works. In each residual block, only the first two convolutional layers are pruned to keep the input and output feature maps to be identical. Hence, only filters from layer 2, 3, 5,... ,9 in stage 1, layer 11, 12,... 18 in stage 2, and layer 23, 24, 26, 27 in stage 3 are pruned.

In the experiment, the well trained neural network models are pruned by both proposed methods and $\ell_1$ are compared from the perspectives of layer-wise pruning, model-wise pruning, filter training behavior, as well as the overall performance. The retraining process is executed on 20 epochs (i.e. 1/8 original training epochs) with a constant learning rate of 0.001.

## 5.2 LAYER-WISE PRUNING ANALYSIS

We demonstrate the effectiveness of our proposed method by comparing the accuracy defection sensitivity of individual layers to the proposed pruning method and $\ell_1$ (Li et al. (2016)) based pruning.

The first row of Fig. 4 shows the layer-wise network accuracy degradation under different pruning ratio according to our method, while the the accuracy degradation caused by the $\ell_1$ based method is demonstrated in the second row.

The following phenomenon is observed: (1) The proposed pruning has a slower accuracy degradation rate compared with the $\ell_1$ ranking based pruning method in majority of layers, especially in the scenario of VGG-16 on the CIFAR-10; (2) In the proposed pruning, the accuracy degradation is slight and degradation rate is small before more than 50% repetitive convolutional filters are pruned; (3) The proposed method can slow down the accuracy degradation of some representative layers of ResNet-32 with the increasing of pruning rate. The reason is the ResNet contains much less filters

Table 1: CIFAR-10 Pruning Configuration

| network | Pruning Rate | FLOPs $(x10^8)$ | FLOPs reduction | prune accuracy | retrain accuracy |
|---|---|---|---|---|---|
| ConvNet | 0 | 8.53 | | | 90.05%* |
| prune A | 40.2% | 5.34 | **37%** | 87.88% | **90.04%** |
| A-L1 | 40.2% | 5.34 | 37% | 83.29% | 89.42% |
| prune B | 64.5% | 2.1 | **63%** | 65.61% | **88.57%** |
| B-L1 | 64.5% | 2.1 | 63% | 53.72% | 87.88% |
| VGG-16 | 0 | 3.31 | | | 90.2%* |
| prune A | 42.9% | 1.85 | **41%** | 88.1% | **90.3%** |
| A-L1 | 42.9% | 1.85 | 41% | 82.4% | 89.82 % |
| prune A | 66.6% | 1.03 | **68%** | 72.1% | **89.9%** |
| B-L1 | 66.6% | 1.03 | 68% | 60.2% | 89.73 % |

*Baseline accuracy

compared to other neural network model. In addition, the deeper layers of ResNet are more sensitive to pruning than the shallower layers, which is different with the other network models.

As such, our proposed method demonstrates significant pruning stability and robustness in the layer-wise pruning. It also means the functionality redundant filters can be more accurately identified and pruned through the proposed interpretive approach.

### 5.3 MODEL-WISE PRUNING ANALYSIS

In this section, we further analyze the sensitivity to pruning of different neural network models. The performance of the proposed method is compared with both the $\ell_1$ and the activation based methods. In the activation based filter pruning method, filters with small output activation are removed Polyak & Wolf (2015)).

Fig. 5 shows the model-wise pruning results of different network models under different pruning rate in the three pruning methods that are named as *Cluster*, *L1*, *Activation* respectively. Here, all convolutional layers in a network model are pruned simultaneously. The results depict that (1) The proposed method outperforms the $\ell_1$ and activation based methods on all models, especially when the neural networks pruning rate is between 20% and 40%; (2) The proposed method has comparable performance with the $\ell_1$ based method when the pruning rate is smaller than 10%.

The above results demonstrate that our proposed method can be applied on full model compression with less accuracy degradation. The overall performance is discussed in the next section.

### 5.4 OVERALL PERFORMANCE EVALUATION

In this section, the overall pruning performance of our proposed method and the $\ell_1$ based method is evaluated and compared, which are depicted as *prune* and *L1* respectively in Table 1 and Table 2. In the evaluation, a retraining procedure is performed to further recover the accuracy. Table 1 shows the

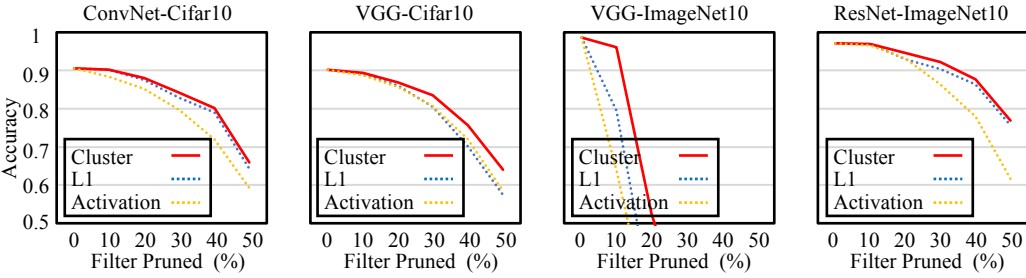

Figure 5: Model-wise pruning analysis.

performance in FLOPs reduction, accuracy after pruning, and accuracy after retraining of ConvNet and VGG on the CIFAR-10 dataset, and Table 2 shows the results of VGG and ResNet-32 on the ImageNet-10 dataset. In the evaluations, two pruning configuration is utilized. In *prune A* and *A-L1*, a small amount of filters are pruned and the original accuracy can be completely recovered by retraining. *prune B* and *B-L1* indicates the scenarios that filters are pruned aggressively to maximize the FLOPs reduction with optimal accuracy. The pruning rate is the percentage of the pruned over the total filters in a neural network model, and the comparison is executed on the same pruning rate.

Table 1 and Table 2 demonstrates that our proposed method introduce less accuracy degradation than the previous $\ell_1$ based method. In addition, the accuracy caused by the proposed pruning can be easier recovered through retraining with better accuracy. For example, in the evaluations of VGG on the CIFAR-10 dataset, the proposed method can achieve 41% FLOPs reduction with retraining accuracy even higher than the original value, however, the accuracy loss caused in the $\ell_1$ based pruning can not be well recovered. As is shown in Table 2, large accuracy loss occurs in the VGG on the ImageNet-10 dataset in the two pruning methods. And the proposed method induces less accuracy degradation and higher retraining accuracy compared with the $\ell_1$ based pruning. Table 2 also shows that it is hard to recover the accuracy drop through retraining in the pruning of the ResNet on the ImageNet-10 dataset, and hence acceptable pruning rate in this scenario is relatively small.

## 5.5 Network Retraining Analysis

In most state-of-the-art works for neural network compression, the retraining is a necessary operation to compensate the inaccurately pruned filters and enhance the network performance. In this section, we evaluate the retraining operation quantitatively and qualitatively.

In Fig. 6, we first quantitatively explore the retraining process in terms of the model accuracy recovery. As shown in Fig. 6, the soiled lines represent the pruned model accuracy recovery based on our proposed method whereas the dot lines represent the $\ell_1$ based method. We can observed that: (1) The models pruned by our method always demonstrate more quickly accuracy recovery. (2) To recover to the same accuracy, our method requires less retraining iterations.

Meanwhile, we also qualitatively explore the retraining process by the AM. Here we mainly focus on the functionality transition during network retraining process. As shown in Fig. 7, we randomly select one filter that has not been pruned by both our method and $\ell_1$ based method and its original function analysis pattern is showed in the first column. Then we use the same visualization methods to visualize the pattern during different retraining iterations, e.g. every 100 iterations. The AM patterns show that, regardless the retraining iterations, the remained filter function of our proposed pruning method remains unchanged. However, the $\ell_1$ based method changes the original filter functionality during retraining process. This means that the retraining process of the $\ell_1$ based method rebuilds the filter functionality, which therefore needs more retraining iterations to restore the model accuracy. The reasons behind that are $\ell_1$ based method may partially destruct the original neural

Table 2: ImageNet-10 Pruning Configuration

| network | Pruning Rate | FLOPs (x$10^{10}$) | FLOPs reduction | prune accuracy | retrain accuracy |
|---|---|---|---|---|---|
| VGG-16 | 0 | 1.54 | | | 98.6%* |
| prune A | 28.3% | 0.84 | **45%** | 51.9% | **96.8%** |
| A-L1 | 28.3% | 0.84 | 45% | 40.7% | 95.7% |
| prune B | 40.9% | 0.57 | **63%** | 25.1% | **94.3%** |
| B-L1 | 40.9% | 0.57 | 63% | 21.2% | 93.2% |
| ResNet-32 | 0 | 2.31 | | | 97.12%* |
| prune A | 21.5% | 1.73 | **25%** | 94.75% | **97.50%** |
| A-L1 | 21.5% | 1.73 | 25% | 92.87% | 96.25% |
| prune B | 30.2% | 1.55 | **33%** | 91.25% | **96.25%** |
| B-L1 | 30.2% | 1.55 | 33% | 90.37% | 94.75 % |

*Baseline accuracy

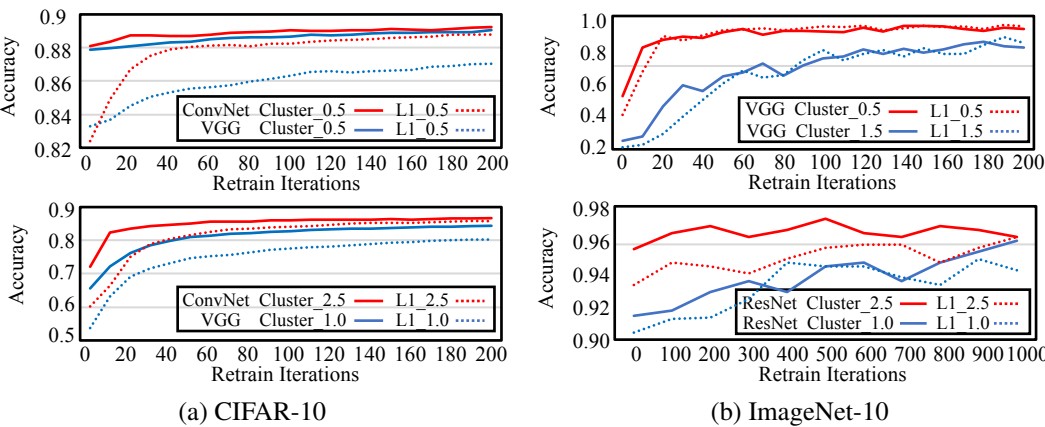

Figure 6: Pruned model accuracy recovery by retraining.

network's functionality composition, which then need more iterations to reconstruct and balance the functionality composition.

That's also the reason why our pruning method causes significantly less accuracy drop and requires less retraining efforts since our pruning method induces less influence to the neural network functionality composition. Therefore, with more interpretable and accurate redundant filter functionality identification and pruning in our method, the costly retraining process will become less necessary, as our retraining analysis results demonstrate.

## 6 CONCLUSION

In this work, through activation maximization based visualization and gradient based filter functionality analysis, we firstly show that convolutional neural network filter redundancy exists in the form of functionality repetition. In other words, the functional repetitive filters could be effectively pruned from neural network to provide computation redundancy. Based on such motivation, in this paper, we propose an interpretable filter pruning method by first clustering filters with same functions together and then reducing the repetitive filters with smallest significance and contribution. Extensive experiments on CIFAR and ImageNet demonstrate the superior performance of our interpretable pruning method over state-of-the-art method, e.g. $\ell_1$ ranking based pruning. By further analyzing the functionality changing of remaining filters in the retraining process, we show that $\ell_1$ ranking based pruning actually partially destructs original neural network's functionality composition. By contrast, our method shows consistent neuron functionality during retraining process, denoting less harm to original network composition. This reveals the underlying reasons of our methods' better performance than L1 method.

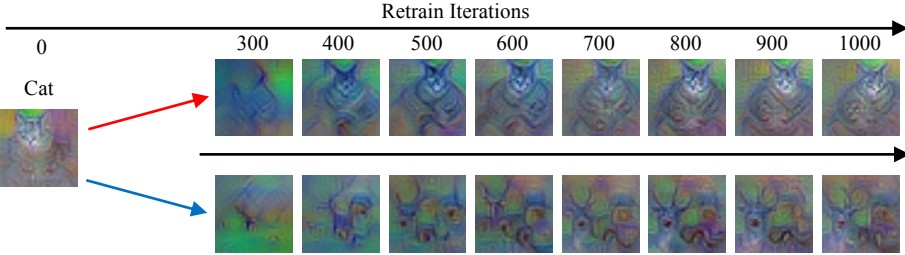

Figure 7: Filter functionality transformation during retraining.

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

## 7 APPENDIX

### 7.1 NETWORK CONFIGURATION

Table 3 and Table 4 show the detailed network structure and pruning configuration of our ConvNet and the VGG-16 on CIFAR10 respectively. The cluster distribution is the total number of clusters in each convolutional layers.

Table 3: ConvNet on CIFAR10 Pruning Configuration

| layer | output | cluster distribution | base filter | prune A filter | A-$\ell_1$ filter | prune B filter | B-$\ell_1$ filter |
|---|---|---|---|---|---|---|---|
| Pr | | | | 0.5 | - | 2.5 | - |
| Conv1_1 | 32x32 | 15 | 64 | 19 | 19 | 36 | 36 |
| Conv2_1 | 16x16 | 20 | 128 | 8 | 8 | 38 | 38 |
| Conv3_1 | 8x8 | 26 | 256 | 21 | 21 | 64 | 64 |
| Conv4_1 | 4x4 | 27 | 512 | 146 | 146 | 329 | 329 |
| Conv5_1 | 2x2 | 46 | 512 | 398 | 398 | 483 | 483 |

Table 4: VGG-16 on CIFAR10 Pruning Configuration

| layer | output | cluster distribution | base filter | prune A filter | A-$\ell_1$ filter | prune B filter | B-$\ell_1$ filter |
|---|---|---|---|---|---|---|---|
| Pr | | | | 0.5 | - | 1 | - |
| Conv1_1 | 32x32 | 15 | 64 | 19 | 19 | 36 | 36 |
| Conv1_2 | 32x32 | 17 | 64 | 3 | 3 | 19 | 19 |
| Conv2_1 | 16x16 | 20 | 128 | 8 | 8 | 38 | 38 |
| Conv2_2 | 16x16 | 31 | 128 | 17 | 17 | 26 | 26 |
| Conv3_1 | 8x8 | 26 | 256 | 21 | 21 | 64 | 64 |
| Conv3_2 | 8x8 | 24 | 256 | 18 | 18 | 64 | 64 |
| Conv3_3 | 8x8 | 14 | 256 | 36 | 36 | 77 | 77 |
| Conv4_1 | 4x4 | 27 | 512 | 146 | 146 | 329 | 329 |
| Conv4_2 | 4x4 | 28 | 512 | 313 | 313 | 424 | 424 |
| Conv4_3 | 4x4 | 38 | 512 | 223 | 223 | 438 | 438 |
| Conv5_1 | 2x2 | 46 | 512 | 398 | 398 | 483 | 483 |
| Conv5_2 | 2x2 | 61 | 512 | 395 | 395 | 432 | 432 |
| Conv5_3 | 2x2 | 40 | 512 | 218 | 218 | 384 | 384 |

### 7.2 FILTER FUNCTIONALITY TRANSITION DURING RETRAINING

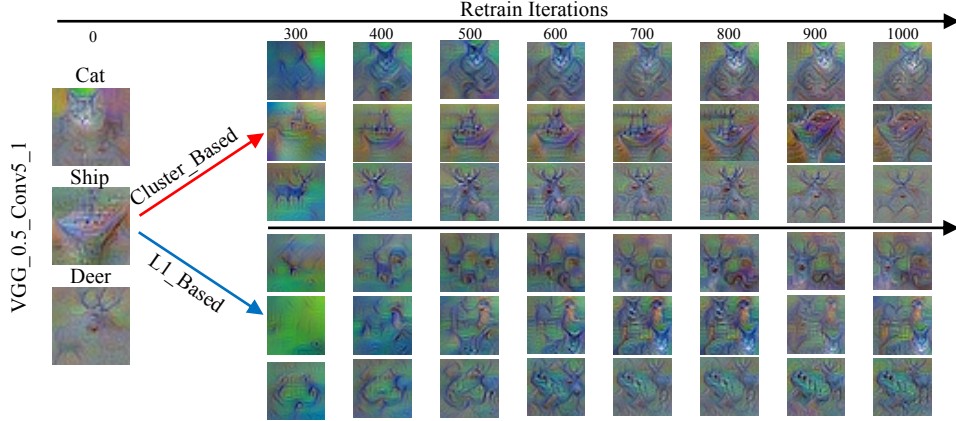

Figure 8: Filter functionality transition during retraining.

