# OpenReview forum: "INTERPRETABLE CONVOLUTIONAL FILTER PRUNING"
_ICLR.cc/2019/Conference_

### Official Review · AnonReviewer1 · 2018-10-28
**No comparisons and claiming something known make it hard to accept this paper**

**Rating:** 3
**Confidence:** 4

**Review:**

This paper claims to have shown some insights about the filters in a neural network. However, it has little contributions that are justifiable to be published and it missed way too many references.

The visualization of filters is hardly any contribution over [1]. The claim that AM is the best visualization tool is a weird statement given that there are many recent references on visualization, such as [2-4], which the authors all missed.

The proposed filter pruning is a simplistic approach that bears little technical novelty, and there has been zero comparison against any filter pruning approach/network compression approach, among the cited references and numerous references that the paper didn't cite, e.g. [5-6]. In this form I cannot accept this paper.

[1] D Bau, B Zhou, A Khosla, A Oliva, and A Torralba. Network Dissection: Quantifying the Intepretability of Deep Visual Representations. In CVPR 2017.
[2] Ramprasaath R. Selvaraju, Michael Cogswell, Abhishek Das, Ramakrishna Vedantam, Devi Parikh Dhruv Batra. Grad-CAM: Visual Explanations from Deep Networks via Gradient-based Localization. ICCV 2017
[3] Jianming Zhang, Zhe Lin, Jonathan Brandt, Xiaohui Shen, Stan Sclaroff. Top-down Neural Attention by Excitation Backprop. ECCV 2016
[4] Ruth Fong and Andrea Vedaldi. Interpretable Explanations of Black Box Algorithms by Meaningful Perturbation. ICCV 2017
[5] Y. Guo, A. Yao and Y. Chen. Dynamic Network Surgery for Efficient DNNs. NIPS 2016
[6] T.-J. Yang, Y.-H. Chen, V. Sze. Designing Energy-Efficient Convolutional Neural Networks using Energy-Aware Pruning. CVPR 2017

---

### Official Review · AnonReviewer3 · 2018-11-02
**Good idea, but needs some improvements.**

**Rating:** 4
**Confidence:** 3

**Review:**

This paper proposes a new method to prune filters of convolutional nets based on a metric which consider functional similarities between filters. Those similarities are computed based on Activation Maximization and gradient information. The proposed method is better than L1 and activation-based methods in terms of accuracy after pruning at the same pruning ratio. The visualization of pruned filters (Fig. 3) shows the effectiveness of the method intuitively.

Overall, the idea in the paper is pretty intuitive and makes sense. The experimental results support the ideas. I think this paper could be accepted if it is improved on the followings:

1. The paper is not very easy to read although the idea is simple.

The equations could be updated and simplified. For example, I'm not sure if S_D in Eq. (3) wants to take V(F_i^(c,l)) and V(F_k^(c,l)) as the arguments. Layer L_l could be just l.

Algorithm 1 is hard to read. At least, one line should correspond to one processing. k is not initialized. It is difficult to understand what each variable represents.

The terms used in Section 4.2 may not be very accurate. First of all, I'm not sure if it is a hierarchical method. It does not perform pruning at multiple levels such as filters, clusters, and layers. Rather, it considers information from multiple levels to determine if a filter should be pruned or not. In that sense, everything is filter level pruning and distinguishing (filter|cluster|layer) level pruning just confuse readers. I'd recommend to simplify the section and describe simply what you do.

2. Comparisons with more recent papers

The proposed method was compared with methods from 2015 and 2016. Model compression is an active area of research and there are a lot of papers. Probably, it makes sense to compare the proposed method against some state-of-the-art methods. Especially, it is interesting to see comparisons against methods with direct optimization of loss function such as (Liu et al. ICCV 2017). We might not need to even consider functionality with such methods.

Liu et al. ICCV 2017: https://arxiv.org/pdf/1708.06519.pdf


* Some other thoughts

** If you look at Figure 3 (a), it looks that there are still a lot of redundant filters. Actually, except the last row, I'm not sure if we can visually find any important difference between (a) and (b). I wonder if the most important thing is that you do not prune unique filters (ones which are not clustered with others). It might be interesting to see a result of the L1-based pruning which does not prunes such filters. If you see an interesting result from that, it could add some value to the paper.

** I'd recommend another proofread.

---

### Official Review · AnonReviewer2 · 2018-11-03
**I think the method proposed in this paper might be reasonable. But I do not suggest acceptance, unless the author can improve the writing and include more experimental results.**

**Rating:** 4
**Confidence:** 4

**Review:**

In this paper, the authors propose a method for pruning the convolutional filters. This method first separates the filters into clusters based on similarities defined with both Activation Maximization (AM) and back-propagation gradients. Then pruning is conducted based on the clustering results, and the contribution index that is calculated based on backward-propagation gradients. The proposed method is compared with a baseline method in the experiments.

I consider the proposed method as novel, since I do not know any filter pruning methods that adopt a similar strategy. Based on my understanding of the proposed method, it might be useful in convolutional filter pruning.

It seems that "interpretable" might not be the most proper word to summarize the method. It looks like that the key concept of this paper, including smilarity defined in Equation (3), and the contribution index defined in Equation (7) are not directly relevant to interpretability. Therefore, I would consider change the title of the paper, for example, to "Convolutional Filter Pruning Based on Functionality ".

In terms of writing, I have difficulty understanding some details about the method.

In filter clustering, how can one run k-means based on pair-wise similarity matrix $S_D$? Do you run kernel k-means, or you  apply PCA to $S_D$ before k-means? What is the criterion of choosing the number of clusters in the process of grid search?

Are filter level pruning, are cluster level pruning and layer level pruning three pruning strategies in the algorithm? It seems to me that you just apply one pruning strategy based on the clusters and contribution index, as shown in Figure 3.

In the subsubsection "Cluster Level Pruning", by "cluster volume size", denoted with$length(C^l_c)$, do you mean the size of cluster, i.e., the number of elements in each cluster? This is the first time I see the term "volume size". I assume the adaptive pruning rate, denoted by $R_{clt}^{(c,l)}$, is a fraction. But it looks to me that $length(C^l_c)$ is an integer. So how can it be true that $R_{clt}^{(c,l)} = length(C^l_c)$?

In the subsubsection "Layer Level Pruning", how is the value of $r$ determined?

The authors have conducted several experiments. These experiments help me understand the advantages of the proposed method. However, in the experiments, the proposed method is compared to only one baseline method. In recent years, a large number of convolutional filter pruning methods have been proposed, as mentioned in the related work section. I am not convinced that the proposed method is one of the best methods among all these existing methods. I would suggest the authors provide more experimental comparison, or explain why comparing with these existing methods is irrelevant.

Since the proposed method is heuristic, I would also like the authors to illustrate that each component of the method is important, via experiment. How would the performance of the proposed method be affected, if we define the similarity $S_D$ in Equation (3) using only $V$ or $\gamma$, rather than both $V$ and $\gamma$? How would the performance of the proposed method be affected, if we prune randomly, rather than prune based on the contribution index?

In summary, I think the method proposed in this paper might be reasonable. But I do not suggest acceptance, unless the author can improve the writing and include more experimental results.

---

### Public Comment · (anonymous) · 2018-11-02
**Sloppy Baseline**

Hi there,
     Filter prune belongs to the structure prune, and you claim in the paper your results are better than previous papers.
However, I don't think so. Lot of papers are shown better performance than yours.
See "Structured Bayesian Pruning via Log-Normal Multiplicative Noise", and "Learning structured sparsity in deep
neural networks".  And there are a lot other papers showing better results than yours.
     From this point, your conclusion is wrong and I don't recommend it for publication since you cannot say you get a new method and then publish. To tell you some tricks, even though at the beginning training stage, I randomly cut some filters and retrain the model, it can say still show better results.

---

> ### Author Response · Authors · 2018-11-02
> **Reply to "Sloppy Baseline" from Authors**
>
> Dear Reviewer:
>
> 1. The comment ignores the problem setup and the contribution details.
> Our work is addressing a totally different situation compared to [1][2].
> In [1][2], they apply sparse constraint during the “training” phase, while our work is to interpret the redundancy of a normally “trained” neural network and propose the functionality oriented pruning method to explore the interpretable neural network optimization. More importantly, our work is proposing a functionality analysis approach with different methods cross-validating each other. We hope such an approach could also be adopted by other compression works to have a better result analysis. Also, the filter L1-ranking based pruning method [3] we are comparing is a well-established work published after [2] in the top conference ICLR 2017, if the authors ignore the problem setup and only chase the final results, we also suggest the reviewer have a discussion with these authors.
>
> 2. Also, we don’t think the reviewer should consider the random pruning as a trick. If the reviewer follows the recent papers closely, you may find that many papers [4][5] discussing the significant redundancy inside neural networks, and different pruning methods (even random pruning) could achieve effectiveness eventually as long as the network is keeping retraining. In other words, there might be a certain optimal network size for a neural network’s functionality, and different pruning methods are just approaching this size. However, the questions of how to interpret the redundancy and what the retraining is doing are rarely addressed. In this work, we interpreted the functionality redundancy in a trained neural network. And our work could effectively and precisely reduce the functionality redundancy with the minimum help of the retraining process. Definitely, we understand why the reviewer favors random pruning so much, in this work we also proved that, functionality wise, the filter L1-ranking based pruning is also a kind of random pruning. Overall, "claiming redundancy" is easy, but "analyzing redundancy" is hard; "random with retraining" is easy, but "precise without retraining" is hard.
>
> Authors.
>
> [1] Structured Bayesian Pruning via Log-Normal Multiplicative Noise. Neklyudov et al., NIPS 2017.
> [2] Learning Structured Sparsity in Deep Neural Networks. Wen et al., NIPS 2016.
> [3] Pruning Filters for Efficient ConvNets. Li et al., ICLR 2017.
> [4] Rethinking the Value of Network Pruning. Liu et al., https://arxiv.org/abs/1810.05270
> [5] Recovering from Random Pruning: On the Plasticity of Deep Convolutional Neural Networks. Mittal et al., WACV 2018.

---

> > ### Public Comment · (anonymous) · 2018-11-03
> > **Not Answer question**
> >
> > I don't it is a different problem setup. You would like to prune the network and finally get your result. Via the method you said why you can prune, is that correct?
> > However, in Structured Bayesian Pruning via Log-Normal Multiplicative Noise, they explain why it can be pruned in the Bayesian method, so how can you say it is different problem setting.
> > In addition, you claim your method is better than previous results and you cannot beat other papers. Even you got a new method, then what is the meaning for that.
> > Again, why don't do a comprehensive comparison and then conclude since you claim "Such an interpretable pruning approach not only offers outstanding computation cost optimization over previous filter pruning methods". I didn't see it offers outstanding computation cost optimization over previous filter pruning methods

---

> > > ### Author Response · Authors · 2018-11-03
> > > **Reply to "Sloppy Baseline" from Authors**
> > >
> > > Dear Reviewer:
> > >
> > > We appreciate that you admitted the novelty of our work. However, we’d like to remind the reviewer again: Firstly, not all the papers with “pruning” in its title have a similar problem setting, “pruning with regularization during training” and “pruning post normal training” are different, and each of them has dedicated publications [1][2]. And we also hope people can explore more settings in different perspectives. Secondly, when we are comparing our research to others, we have already clearly shown our advantage over the baselines, and hope you can also carefully read our advantage in the retraining part.
> > >
> > > Authors.
> > >
> > > [1] Fast convnets using group-wise brain damage. Lebedev et al., CVPR 2016.
> > > [2] Network trimming: A data-driven neuron pruning approach towards efficient deep architectures. Hu et al., 2016. https://arxiv.org/abs/1607.03250

---

> > > > ### Public Comment · (anonymous) · 2018-11-03
> > > > **Reply to "Sloppy Baseline"  (OP May not be a reviewer)**
> > > >
> > > >  It is very likely that the OP for this thread is not a reviewer.
> > > >
> > > >  Bare in mind anyone can sign as anonymous, whilst reviewers have been signing as anonreviewer\d+ .
> > > >
> > > >  Furthermore the comments that the OP has written are very hard to parse, there was little effort put in to proof-checking the grammar.  If OP is indeed a reviewer then they should probably conform to the standards and sign as anonreviewer\d+ .
> > > >
> > > >  Openreview should probably restrict the comments from non-reviewers. I feel this is creating a lot of clutter and turning this process/platform in to some form of social medium.

---

> > > > > ### Author Response · Authors · 2018-11-03
> > > > > **Reply to "OP May not be a reviewer" from Authors**
> > > > >
> > > > > Dear Another Reviewer in this thread:
> > > > >
> > > > > Thank you so much for your fair comment in this thread.
> > > > >
> > > > > We are trying to collect all the feedback and interact with all the readers since we are taking the OpenReview as a very serious academic society rather social medium.  That's why we are doing our best to reply to the OP with detailed explanation and references.
> > > > >
> > > > > Therefore, we also agree with you to some extent, since we are always hoping the OP can raise more constructive questions and help us to improve.
> > > > >
> > > > > Again, thank you so much for your support.
> > > > >
> > > > > Authors

---

> > > > ### Public Comment · (anonymous) · 2018-11-03
> > > > **Still Not Answer My Question**
> > > >
> > > > First of all, you didn't compare with previous published showing good results and didn't explain why you don't compare with them. In addition, you didn't explain the difference between them. Since “pruning with regularization during training” and “pruning post normal training” is different, and why people choose to do this, and what is reason behind them. If the final goal is to prune the network, and accelerate the network, so why you still claim there are different?
> > > > Second, you claim baseline accuracy of cifar10 under Vgg16 is 90.2%, and you got 90.3%, then I am telling you the baseline is around 93%. I don't have to search a lot, just randomly search on github, https://github.com/geifmany/cifar-vgg. And they got 93%, how do you explain this. From this way, your accuracy has decreased 3% and lots of papers do the pruning without the accuracy decrease, so how can you explain the advantage of your method.

---

> > > > > ### Public Comment · (anonymous) · 2018-11-03
> > > > > **Please watch your tone when commenting others work**
> > > > >
> > > > > I am not an author of this work and I am not an expert in this field. But I really dislike your tone when you comment on others' work. Your comments are really unconvincing.
> > > > > Firstly, you mention there are good results from others publication, but you don't list any publications to support your argument, whereas the response of the authors referred some of the literature.
> > > > > Secondly, the link you mention to achieve 93% accuracy did not work. You should check that and give concrete papers.
> > > > > Thirdly, Please avoid using questions like ' how do you explain this' and so 'why you still claim there are different'. These are very offensive, this is not social media but an academic venue.
> > > > > Finally, I recommend that the program committees of the ICLR conference should consider restricting the comments from non-reviewers. The authors have to waste much unnecessary time responding to low-quality comments here. Thank you very much.

---

> > > > > > ### Public Comment · (anonymous) · 2018-11-03
> > > > > > **See concrete paper about real cifar10 accuracy by vgg16**
> > > > > >
> > > > > > 1. "Long Live TIME: Improving Lifetime for Training-In-Memory
> > > > > > Engines by Structured Gradient Sparsification".    This paper shows 92.5%
> > > > > > 2. Online Filter Clustering and Pruning for Efficient Convnets
> > > > > >  This paper shows 93.25%.
> > > > > > 3. Learning Efficient Convolutional Networks through Network Slimming.
> > > > > > This paper shows 93.66%
> > > > > >
> > > > > > Now I show the baseline is much better than the baseline you choose as 90.2%. So consider changing the conclusion of your paper?

---

> > > > > > > ### Author Response · Authors · 2018-11-03
> > > > > > > **Reply to "Sloppy Baseline" from Authors**
> > > > > > >
> > > > > > > Dear Reviewer:
> > > > > > >
> > > > > > > First of all, we think we have already answered the problem setting. “pruning with regularization during training” and “pruning post normal training” are the most intuitive explanation we can provide. For more details, please refer to the paper [1], which is published in AAAI 2018.
> > > > > > >
> > > > > > > Secondly, here is the answer regarding the baseline difference. It’s common that the baseline variance of the same model exists between different works [1][2][3], since people usually train published models from scratch for convenience. We did the same in our work.
> > > > > > > However, we didn’t put much effort into chasing the highest performance of the original method, since that’s not the major focus of our work. And this difference actually doesn’t defect our findings of filter functionality analysis, functionality redundancy elimination, retraining analysis, etc. However, we can definitely improve the baseline in a future version.
> > > > > > >
> > > > > > > Again, we sincerely ask the reviewer to pay more attention to our methods and contributions in our work and other referenced ones, rather than chasing results regardless of problem settings and perfecting baselines. Otherwise, this is an issue of our research philosophy difference, which can’t be well resolved.
> > > > > > >
> > > > > > > Authors.
> > > > > > >
> > > > > > > [1] Auto-balanced Filter Pruning for Efficient Convolutional neural networks. Ding et al., AAAI 2018.
> > > > > > > [2] Pruning Filters for Efficient ConvNets. Li et al., ICLR 2017.
> > > > > > > [3] Learning to Prune Filters in Convolutional Neural Networks. Huang et al., WAVC2018.

---

> > > > > > > > ### Public Comment · (anonymous) · 2018-11-03
> > > > > > > > **Thank you for the reply**
> > > > > > > >
> > > > > > > > I appreciate the efforts authors made to address the comments by others. I think since some commenter and authors are not on the same page,  reviewers should not be influenced by these comments and judge on their own. Thank you.

---

> > > > > > > > ### Public Comment · (anonymous) · 2018-11-03
> > > > > > > > **some thoughts**
> > > > > > > >
> > > > > > > > Hi Authors,
> > > > > > > >
> > > > > > > > Thanks for the continuing effort on clarifying your paper. In the end, unfortunately I don't feel the argument you gave regarding “pruning with regularization during training” and “pruning post normal training” is convincing. As the person pointed out, if the goal is to prune the network, and accelerate the network, I do not see there is any reason people do not go for the approach that achieves the best results regardless if it falls into the category of pruning with regularization during training or pruning post normal training. In other words, it would be helpful if you can explain your approach addresses some of the limitations/issues of [1] despite being less accurate. Hope this makes some sense.
> > > > > > > >
> > > > > > > >
> > > > > > > >
> > > > > > > > [1] 3. Learning Efficient Convolutional Networks through Network Slimming.

---

> > > > > > > > > ### Author Response · Authors · 2018-11-04
> > > > > > > > > **Reply to "some thoughts" from Authors**
> > > > > > > > >
> > > > > > > > > Dear Reviewer,
> > > > > > > > >
> > > > > > > > > Thanks for your comment.
> > > > > > > > >
> > > > > > > > > 1. The “pruning with regularization during training” and “pruning post normal training” are clearly divided into two different categories and have been well discussed in [1]. Post design optimization is a well-recognized concept in many research areas. And there are also many excellent works emerging for such a pruning approach [2][3]. For more details, we recommend reviewers to refer to these papers. Overall, rather than judging which is better, these are two complementary approaches.
> > > > > > > > >
> > > > > > > > > 2. We hope the reviewer can broaden the understanding of pruning. As we mentioned in our first reply, different pruning methods are just approaching the minimal network size [4][5]. It’s more important to understand the neural network with pruning. Our contribution in this work is not only pruning, but also interpreting the source of network redundancy. And based on this analysis, we proposed the method to effectively and precisely reduce the functionality redundancy.
> > > > > > > > >
> > > > > > > > > Authors.
> > > > > > > > >
> > > > > > > > > [1] Auto-balanced Filter Pruning for Efficient Convolutional neural networks. Ding et al., AAAI 2018.
> > > > > > > > > [2] NetAdapt: Platform-Aware Neural Network Adaptation for Mobile Applications. Yang et al., ECCV 2018.
> > > > > > > > > [3] ThiNet: A Filter Level Pruning Method for Deep Neural Network Compression. Luo et al. ICCV, 2017.
> > > > > > > > > [4] Rethinking the Value of Network Pruning. Liu et al., https://arxiv.org/abs/1810.05270
> > > > > > > > > [5] Learning Efficient Convolutional Networks through Network Slimming. Liu et al., ICCV 2017.

---

> ### Author Response · Authors · 2018-11-04
> **To All Reviewers from Authors**
>
> Dear Reviewers:
>
> We have done our best to clarify our works to the original poster.
> If you are looking for answers regarding the question of "problem settings of pruning trained models" and "baseline selection", please refer to the below replies.
>
> We are still very open to other questions, and we will do our best to reply to those constructive ones.
> However, we hope future reviewers could fully read our paper and fairly review our contributions without being influenced by some very aggressive comments below.
>
> Authors.

---

### Public Comment · (anonymous) · 2018-11-04
**Comments are just comments, not reviews.**

I appreciate the efforts authors made to address the comments by others. I think since some comments below are very aggressive and annoying,  so I suggest that all reviewers should judge this paper fairly and independently. Thank you for your understanding.

---

### Meta-Review · Area_Chair1 · 2018-12-16
**Unanimous rejection.**

**Confidence:** 4
**Recommendation:** Reject

**Metareview:**

The current version of the paper receives a unanimous rejection from reviewers, as the final proposal.